# Training shapes the curvature
# of shallow neural network representations

**Jacob A. Zavatone-Veth**[*]                                    JZAVATONEVETH@G.HARVARD.EDU
*Department of Physics and Center for Brain Science, Harvard University, Cambridge, MA, USA*
**Julian A. Rubinfien**[*]                                    JULIAN.RUBINFIEN@YALE.EDU
*Department of Physics, Yale University, New Haven, CT, USA*
**Cengiz Pehlevan**                                    CPEHLEVAN@SEAS.HARVARD.EDU
*John A. Paulson School of Engineering and Applied Sciences and Center for Brain Science,*
*Harvard University, Cambridge, MA, USA*

**Editors:** Sophia Sanborn, Christian Shewmake, Simone Azeglio, Arianna Di Bernardo, Nina Miolane

## Abstract

We study how training shapes the Riemannian geometry induced by neural network feature maps. At infinite width, shallow neural networks induce highly symmetric metrics on input space. Feature learning in networks trained to perform simple classification tasks magnifies local areas and reduces curvature along decision boundaries. These changes are consistent with previously proposed geometric approaches for hand-tuning of kernel methods to improve generalization.

## 1. Introduction

In a series of influential papers, Amari and Wu proposed that one could improve the generalization performance of support vector machine (SVM) classifiers through data-dependent transformations of the kernel to expand the Riemannian volume element near decision boundaries (Amari and Wu, 1999; Wu and Amari, 2002; Williams et al., 2007). This proposal was based on the idea that this local magnification of areas improves the discriminability of classes in input space (Cho and Saul, 2011; Amari and Wu, 1999; Burges, 1999). Over the past decade, SVMs have largely been eclipsed by neural networks, whose ability to flexibly learn features from data is believed to underlie their superior generalization performance (LeCun et al., 2015; Zhang et al., 2021). Previous works have explored some aspects of the Riemannian geometry induced by neural networks with random parameters (Poole et al., 2016; Amari et al., 2019; Cho and Saul, 2009, 2011; Zavatone-Veth and Pehlevan, 2022), but have not characterized data-dependent changes in representational geometry over training.

In this work, we explore the possibility that neural networks learn to enhance local input discriminability automatically over the course of training. We first analytically compute the curvature of the metric induced by infinitely wide shallow networks with Gaussian weights and smooth activation functions, showing that it is highly symmetric. We then empirically show that training on simple classification tasks expands the volume element and reduces the curvature along decision boundaries, largely consistent with the hand-engineered modifications proposed by Amari and Wu. In total, our results provide a preliminary picture of how feature learning shapes local input discriminability.

---

[*] JAZV and JAR contributed equally to this work.

## 2. Preliminaries

Consider a shallow neural network with $d$-dimensional inputs, and an $n$-dimensional feature space given by a feature map $\mathbf{\Phi}$ of the form

$$\Phi_j(\mathbf{x}) = \frac{1}{\sqrt{n}}\phi(\mathbf{w}_j \cdot \mathbf{x} + b_j) \tag{1}$$

for weights $\mathbf{w}_j$, biases $b_j$, and an activation function $\phi$. Here, we index input space dimensions by $\mu, \nu, \rho, \ldots = 1, \ldots, d$ and hidden layer dimensions by $i, j, k, \ldots = 1, \ldots, n$, and denote the Euclidean inner product by $\mathbf{w}_j \cdot \mathbf{x} = w_{j\mu}x_\mu$. We adopt Neural Tangent Kernel (NTK) parameterization for the feature map—i.e., we include a factor of $n^{-1/2}$ in its definition—to ensure that the infinite-width limit $n \to \infty$ is well-defined (Jacot et al., 2018; Yang and Hu, 2021). Our subsequent results will not assume a particular form for the readout layer following this feature map. The remainder of the network's architecture will affect the training dynamics of the feature map parameters while leaving its functional form unchanged, hence some of our results carry over to the first hidden layer of a deep network.

We will always assume that $n \geq d$, such that the feature map does not compress the dimensionality of the inputs. Then, if the activation function $\phi$ is $k$ times continuously differentiable and the weight vectors are linearly independent, the image of some subset of input space under the feature map is a $d$-dimensional $\mathcal{C}^k$ manifold $\mathcal{M}$ embedded in $\mathbb{R}^n$ (Dodson and Poston, 1991). We will always assume that $k \geq 3$, and will generally assume $k = \infty$, such that we have a smooth manifold. Then, the Euclidean metric on $\mathbb{R}^n$ induces a metric

$$g_{\mu\nu}(\mathbf{x}) = \partial_\mu \Phi_i \partial_\nu \Phi_i = \frac{1}{n}\phi'(\mathbf{w}_j \cdot \mathbf{x} + b_j)^2 w_{j\mu}w_{j\nu} \tag{2}$$

on the submanifold $\mathcal{M}$, where we write $\partial_\mu \equiv \partial/\partial x^\mu$ and denote by $\phi'$ the first derivative of $\phi$ with respect to its argument. For later results, we will also assume that $\phi$ is exponentially bounded at infinity.

We will consider two characteristics of the Riemannian geometry of this manifold. First, the volume element on $\mathcal{M}$ is given by

$$dV = \sqrt{\det g}\, d^d x, \tag{3}$$

where the factor $\sqrt{\det g}$ measures how local areas in input space are magnified by the feature map (Dodson and Poston, 1991; Amari and Wu, 1999; Burges, 1999). Second, we consider the intrinsic curvature of the manifold, which is completely characterized by the Riemann tensor $R_{\mu\nu\alpha\beta}$ (Dodson and Poston, 1991). For a metric of the form (2), we show in Appendix A that the Riemann tensor can be expressed in the particularly simple form

$$R_{\mu\nu\alpha\beta} = -\frac{3}{4}g^{\rho\lambda}(\partial_\rho g_{\mu\alpha}\partial_\lambda g_{\nu\beta} - \partial_\rho g_{\mu\beta}\partial_\lambda g_{\nu\alpha}) \tag{4}$$

thanks to the symmetry of $\partial_\alpha g_{\mu\nu}$ under permutation of its indices. As a tractable measure, we focus on the Ricci curvature scalar $R = g^{\beta\nu}R^\alpha_{\nu\alpha\beta}$, which measures the deviation of the volume of an infinitesimal geodesic ball in the manifold from that in flat space.

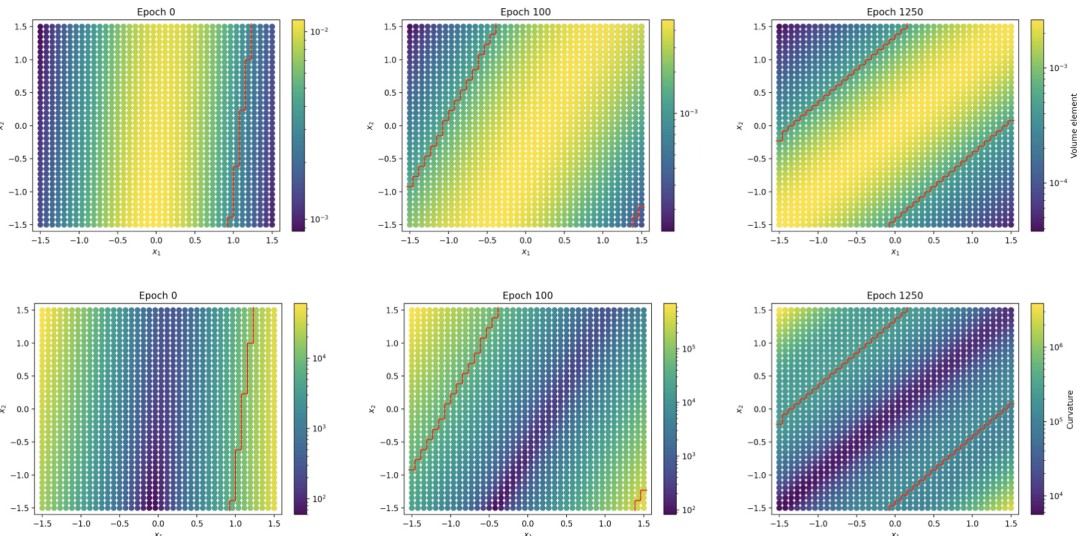

Figure 1: Evolution of the volume element (*top*) and scalar curvature (*bottom*) over training in a network trained to perform an XOR classification task. Red lines indicate the decision boundaries of the network. See Appendix C for experimental details.

## 3. Representational geometry of infinite-width shallow networks

We first characterize the metric induced by infinite-width networks ($n \to \infty$). For Gaussian weights and biases $\mathbf{w}_j \sim \mathcal{N}(\mathbf{0}, \sigma^2 \mathbf{I}_d)$ and $b_j \sim \mathcal{N}(0, \zeta^2)$, the hidden layer representation at infinite width is described by the neural network Gaussian process (NNGP) kernel (Neal, 1996; Williams, 1997; Matthews et al., 2018; Lee et al., 2018):

$$k(\mathbf{x}, \mathbf{y}) = \lim_{n \to \infty} \frac{1}{n} \mathbf{\Phi}(\mathbf{x}) \cdot \mathbf{\Phi}(\mathbf{y}) = \mathbb{E}_{\mathbf{w} \sim \mathcal{N}(\mathbf{0}, \sigma^2 \mathbf{I}_d), b \sim \mathcal{N}(0, \zeta^2)}[\phi(\mathbf{w} \cdot \mathbf{x} + b)\phi(\mathbf{w} \cdot \mathbf{y} + b)], \qquad (5)$$

where the limit is almost-sure. As infinite networks with NTK parameterization do not learn features, this kernel completely describes the representation even after training, including in deep networks with additional hidden layers following this feature map (Jacot et al., 2018; Yang and Hu, 2021).

In Appendix B, we use the results of Burges (1999) to show that this kernel induces a metric

$$g_{\mu\nu} = \mathbb{E}_{\mathbf{w} \sim \mathcal{N}(\mathbf{0}, \sigma^2 \mathbf{I}_d), b \sim \mathcal{N}(0, \zeta^2)}[\phi'(\mathbf{w} \cdot \mathbf{x} + b)^2 w_\mu w_\nu]. \qquad (6)$$

This is simply the parameter average of the finite-width metric (2), which by the strong law of large numbers is its almost-sure infinite-width limit. This metric can be written in the more illuminating form

$$g_{\mu\nu} = e^{\Omega(\|\mathbf{x}\|^2)}[\delta_{\mu\nu} + 2\Omega'(\|\mathbf{x}\|^2)x_\mu x_\nu], \qquad (7)$$

where the function $\Omega$ is defined via

$$e^{\Omega(\|\mathbf{x}\|^2)} = \sigma^2 \mathbb{E}_{z \sim \mathcal{N}(0, \sigma^2 \|\mathbf{x}\|^2 + \zeta^2)}[\phi'(z)^2]. \qquad (8)$$

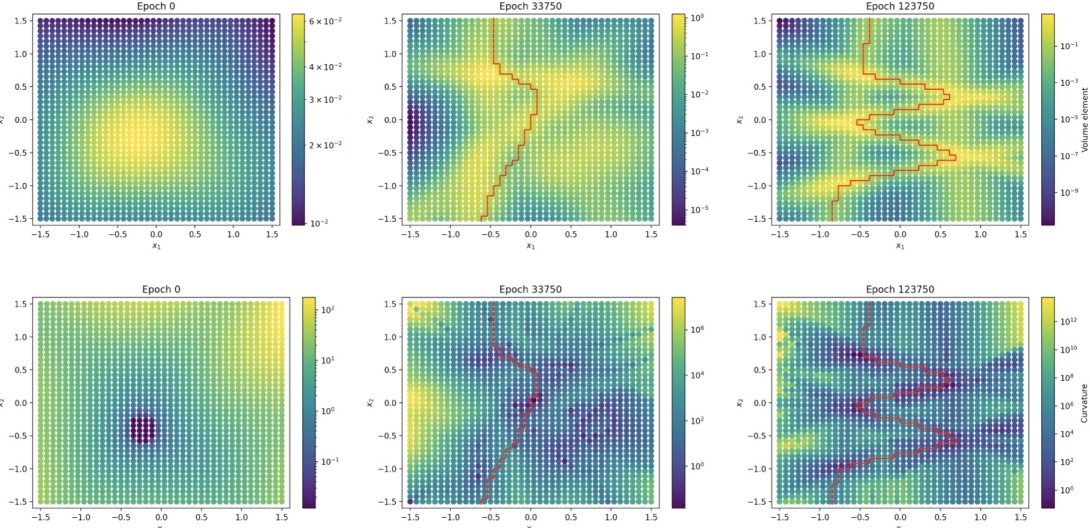

Figure 2: Evolution of the volume element (*top*) and scalar curvature (*bottom*) over training in a network trained to classify points separated by a sinusoidal boundary. Red lines indicate the decision boundaries of the network. See Appendix C for details.

Such metrics have det $g = e^{d\Omega}(1 + 2\|\mathbf{x}\|^2\Omega')$ and Ricci scalar

$$R = -\frac{3(d-1)e^{-\Omega}(\Omega')^2\|\mathbf{x}\|^2}{(1 + 2\|\mathbf{x}\|^2\Omega')^2}\left[d + 2 + 2\|\mathbf{x}\|^2\left((d-2)\Omega' + 2\frac{\Omega''}{\Omega'}\right)\right]. \qquad (9)$$

Thus, all curvature quantities are spherically symmetric, depending only on $\|\mathbf{x}\|^2$. This generalizes the results of Cho and Saul (2011) for threshold-power law functions to arbitrary smooth activation functions. In Appendix B, we explicitly evaluate the curvature quantities for certain activation functions, such as the Gauss error function, for which the required integrals are tractable.

## 4. Changes in geometry during training

We finally consider how the curvature of the induced metric changes during training. Changes in curvature during gradient descent training are challenging to study analytically, because solvable models—deep linear networks (Saxe et al., 2013)—trivially yield flat metrics with constant magnification factors. Therefore, we resort to numerical experiments. In Figures 1 and 2, we show how the volume element and Ricci scalar change over training in networks trained to perform XOR and sinusoid classification tasks, respectively. For a given draw of the initial parameters at finite width, the curvature quantities are not exactly symmetric. As training progresses, we find that the volume element induced by the network grows relatively large — and the scalar curvature relatively small — in the vicinity of the decision boundary.

In Figure 3, we provide preliminary evidence that a similar phenomenon may be present in networks trained to classify MNIST images. For visualization purposes, we plot a regularized volume element at synthetic images generated by linearly interpolating between two input

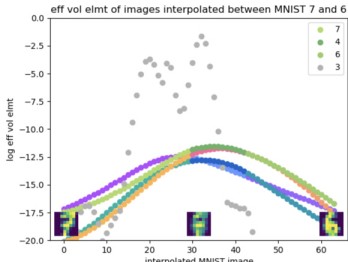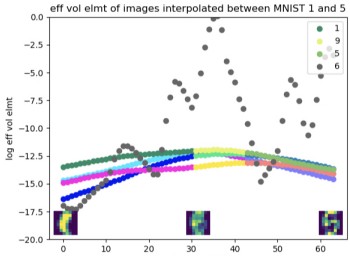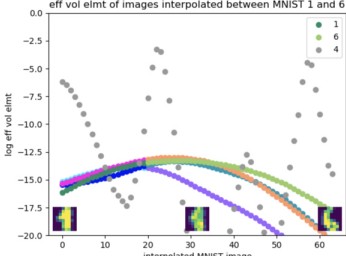

Figure 3: Effective volume element induced at interpolated images by networks trained to classify MNIST digits. Each colored line is a replicate network: the abscissa is the interpolated MNIST image; the ordinate is the log of an effective volume element; the color is the network's predicted digit for that image. Black/gray lines show negative controls (untrained networks). Lower left and right inset images are the initial and final images of the interpolation, respectively; the middle inset image is the interpolated image at which the network's prediction first changes.

images (see Appendix C for details). We find that this volume element is consistently large at interpolated images near the network's decision boundary.

## 5. Conclusions

To conclude, we have shown that training on simple tasks shapes the Riemannian geometry induced by neural network representations by magnifying area and suppressing curvature along decision boundaries. The magnification of areas is consistent with the proposal of Amari and Wu (Amari and Wu, 1999; Wu and Amari, 2002; Williams et al., 2007). In future work, it will be interesting to investigate whether these phenomena are visible in deep networks trained to perform more complex tasks, and to investigate the links between induced geometry and generalization.

## Acknowledgments

We thank Sheng Yang for helpful discussions, and Blake Bordelon for helpful comments on our manuscript. JAZ-V and CP were supported by a Google Faculty Research Award and NSF DMS-2134157.

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

## Appendix A. Simplification of the Riemann tensor for a general shallow network

In this section, we show how the general form of the Riemann tensor can be simplified for metrics of the form considered here. As elsewhere, our conventions follow Dodson and Poston (1991). For a metric of the general form

$$g_{\mu\nu} = \mathbb{E}_{\mathbf{w},b}[\phi'(\mathbf{w} \cdot \mathbf{x} + b)^2 w_\mu w_\nu], \tag{A.1}$$

we have

$$\partial_\alpha g_{\mu\nu} = 2\mathbb{E}_{\mathbf{w},b}[\phi'(\mathbf{w} \cdot \mathbf{x} + b)\phi''(\mathbf{w} \cdot \mathbf{x} + b)w_\alpha w_\mu w_\nu], \tag{A.2}$$

which is symmetric under permutation of its indices. Therefore, the Christoffel symbols of the second kind reduce to

$$\Gamma^\alpha_{\beta\gamma} = \frac{1}{2}g^{\alpha\mu}(\partial_\beta g_{\gamma\mu} - \partial_\mu g_{\beta\gamma} + \partial_\gamma g_{\mu\beta}) \tag{A.3}$$

$$= \frac{1}{2}g^{\alpha\mu}\partial_\beta g_{\gamma\mu}. \tag{A.4}$$

The $(3, 1)$ Riemann tensor is then

$$R^\mu_{\nu\alpha\beta} = \partial_\alpha\Gamma^\mu_{\beta\nu} - \partial_\beta\Gamma^\mu_{\alpha\nu} + \Gamma^\rho_{\alpha\nu}\Gamma^\mu_{\beta\rho} - \Gamma^\rho_{\beta\nu}\Gamma^\mu_{\alpha\rho} \tag{A.5}$$

$$= \frac{1}{2}\left[\partial_\alpha(g^{\mu\rho}\partial_\beta g_{\nu\rho}) - \partial_\beta(g^{\mu\rho}\partial_\alpha g_{\nu\rho})\right]$$
$$+ \frac{1}{4}\left[(g^{\rho\lambda}\partial_\alpha g_{\nu\lambda})(g^{\mu\sigma}\partial_\beta g_{\rho\sigma}) - (g^{\rho\lambda}\partial_\beta g_{\nu\lambda})(g^{\mu\sigma}\partial_\alpha g_{\rho\sigma})\right] \tag{A.6}$$

$$= \frac{1}{2}\left[\partial_\alpha g^{\mu\rho}\partial_\beta g_{\nu\rho} - \partial_\beta g^{\mu\rho}\partial_\alpha g_{\nu\rho} + g^{\mu\rho}(\partial_\alpha\partial_\beta g_{\nu\rho} - \partial_\beta\partial_\alpha g_{\nu\rho})\right]$$
$$+ \frac{1}{4}\left[-\partial_\alpha g_{\nu\lambda}\partial_\beta g^{\mu\lambda} + \partial_\beta g_{\nu\lambda}\partial_\alpha g^{\mu\lambda}\right] \tag{A.7}$$

$$= \frac{3}{4}(\partial_\alpha g^{\mu\rho}\partial_\beta g_{\nu\rho} - \partial_\beta g^{\mu\rho}\partial_\alpha g_{\nu\rho}), \tag{A.8}$$

where we have used the fact that partial derivatives commute and recalled the matrix calculus identity

$$\partial_\alpha g^{\mu\nu} = -g^{\mu\rho}g^{\nu\lambda}\partial_\alpha g_{\rho\lambda}. \tag{A.9}$$

Then, the $(4, 0)$ Riemann tensor is

$$R_{\mu\nu\alpha\beta} = g_{\mu\lambda}R^\lambda_{\nu\alpha\beta} \tag{A.10}$$

$$= -\frac{3}{4}g^{\rho\lambda}(\partial_\alpha g_{\mu\rho}\partial_\beta g_{\nu\lambda} - \partial_\beta g_{\mu\rho}\partial_\alpha g_{\nu\lambda}) \tag{A.11}$$

which, given the permutation symmetry of the derivatives of the metric, can be re-expressed as

$$R_{\mu\nu\alpha\beta} = -\frac{3}{4}g^{\rho\lambda}(\partial_\rho g_{\mu\alpha}\partial_\lambda g_{\nu\beta} - \partial_\rho g_{\mu\beta}\partial_\lambda g_{\nu\alpha}). \tag{A.12}$$

It is then easy to see that the simplified formula for the Riemann tensor has the expected symmetry properties under index permutation:

$$R_{\mu\nu\alpha\beta} = -R_{\mu\nu\beta\alpha} \tag{A.13}$$

$$R_{\mu\nu\alpha\beta} = -R_{\nu\mu\alpha\beta} \tag{A.14}$$

$$R_{\mu\nu\alpha\beta} = +R_{\alpha\beta\mu\nu} \tag{A.15}$$

and satisfies the Bianchi identity

$$R_{\mu\nu\alpha\beta} + R_{\mu\alpha\beta\nu} + R_{\mu\beta\nu\alpha} = 0. \tag{A.16}$$

Finally, the Ricci scalar is

$$R = g^{\beta\nu} R^{\alpha}_{\nu\alpha\beta} \tag{A.17}$$

$$= -\frac{3}{4} g^{\mu\alpha} g^{\nu\beta} g^{\rho\lambda} (\partial_\alpha g_{\mu\rho} \partial_\beta g_{\nu\lambda} - \partial_\beta g_{\mu\rho} \partial_\alpha g_{\nu\lambda}) \tag{A.18}$$

$$= -\frac{3}{4} g_{\rho\lambda} (\partial_\alpha g^{\alpha\rho} \partial_\beta g^{\beta\lambda} - \partial_\beta g^{\alpha\rho} \partial_\alpha g^{\beta\lambda}). \tag{A.19}$$

## Appendix B. Derivation of curvature quantities at infinite width

In this section, we derive the curvature quantities for the infinite-width metric (or, equivalently, the average finite-width metric) at initialization:

$$g_{\mu\nu} = \mathbb{E}_{\mathbf{w}\sim\mathcal{N}(\mathbf{0},\sigma^2\mathbf{I}_d), b\sim\mathcal{N}(0,\zeta^2)}[\phi'(\mathbf{w}\cdot\mathbf{x}+b)^2 w_\mu w_\nu]. \tag{B.1}$$

For the remainder of this section, we will simply write the expectation over $\mathbf{w} \sim \mathcal{N}(\mathbf{0}, \sigma^2\mathbf{I}_d)$ and $b \sim \mathcal{N}(0, \zeta^2)$ as $\mathbb{E}[\cdot]$. We let

$$z \equiv \mathbf{w}\cdot\mathbf{x} + b, \tag{B.2}$$

which has an induced $\mathcal{N}(0, \sigma^2\|\mathbf{x}\|^2 + \zeta^2)$ distribution. We remark that it is easy to show that (B.1) is the metric induced by the NNGP kernel

$$k(\mathbf{x}, \mathbf{y}) = \mathbb{E}[\phi(\mathbf{w}\cdot\mathbf{x}+b)\phi(\mathbf{w}\cdot\mathbf{y}+b)] \tag{B.3}$$

using the formula (Burges, 1999)

$$g_{\mu\nu} = \frac{1}{2}\frac{\partial^2}{\partial x_\mu \partial x_\nu}k(\mathbf{x}, \mathbf{x}) - \left[\frac{\partial^2}{\partial y_\mu \partial y_\nu}k(\mathbf{x}, \mathbf{y})\right]_{\mathbf{y}=\mathbf{x}} \tag{B.4}$$

for a sufficiently smooth activation function.

Applying Stein's lemma twice, we have

$$g_{\mu\nu} = \mathbb{E}[\phi'(z)^2 w_\mu w_\nu] \tag{B.5}$$

$$= \sigma^2\mathbb{E}[\phi'(z)^2]\delta_{\mu\nu} + 2\sigma^2\mathbb{E}[\phi'(z)\phi''(z)w_\nu]x_\mu \tag{B.6}$$

$$= \sigma^2\mathbb{E}[\phi'(z)^2]\delta_{\mu\nu} + 2\sigma^4\mathbb{E}[\phi''(z)^2 + \phi'(z)\phi'''(z)]x_\mu x_\nu. \tag{B.7}$$

Then, we can see that the metric is of a special form. Noting that $\mathbb{E}[\phi'(z)^2] \geq 0$ and that

$$\sigma^2 \mathbb{E}[\phi''(z)^2 + \phi'(z)\phi'''(z)] = \sigma^2 \frac{d}{d(\sigma^2\|\mathbf{x}\|^2 + \zeta^2)} \mathbb{E}[\phi'(z)^2] \tag{B.8}$$

$$= \frac{d}{d\|\mathbf{x}\|^2} \mathbb{E}[\phi'(z)^2] \tag{B.9}$$

by Price's theorem (Price, 1958) and the chain rule, we may write

$$g_{\mu\nu} = e^{\Omega(\|\mathbf{x}\|^2)}[\delta_{\mu\nu} + 2\Omega'(\|\mathbf{x}\|^2)x_\mu x_\nu], \tag{B.10}$$

where we have defined the function $\Omega(\|\mathbf{x}\|^2)$ by

$$\exp\Omega(\|\mathbf{x}\|^2) \equiv \sigma^2 \mathbb{E}[\phi'(z)^2]. \tag{B.11}$$

## B.1. Curvature quantities for metrics of the form induced by the shallow NNGP kernel

Motivated by the metric induced by the shallow NNGP kernel, we consider metrics of the general form

$$g_{\mu\nu} = e^{\Omega(\|\mathbf{x}\|^2)}[\delta_{\mu\nu} + 2\Omega'(\|\mathbf{x}\|^2)x_\mu x_\nu], \tag{B.12}$$

where $\Omega$ is a smooth function with derivative $\Omega'$. For brevity, we will henceforth suppress the argument of $\Omega$.

Such metrics have determinant

$$\det g = e^{d\Omega}(1 + 2\|\mathbf{x}\|^2\Omega') \tag{B.13}$$

by the matrix determinant lemma, and inverse

$$g^{\mu\nu} = e^{-\Omega}\left[\delta_{\mu\nu} - \frac{2\Omega'}{1 + 2\|\mathbf{x}\|^2\Omega'}x_\mu x_\nu\right] \tag{B.14}$$

by the Sherman-Morrison formula. It is also easy to see that the eigenvalues of the metric at any given point $\mathbf{x}$ are $e^{\Omega}(1 + 2\|\mathbf{x}\|^2\Omega')$ with corresponding eigenvector $\mathbf{x}/\|\mathbf{x}\|$, and $e^{\Omega}$ with multiplicity $d - 1$, with eigenvectors lying in the null space of $\mathbf{x}$.

We now consider the Riemann tensor. For such metrics, we have

$$\partial_\alpha g_{\mu\nu} = 2e^{\Omega}\Omega'(x_\alpha\delta_{\mu\nu} + x_\mu\delta_{\alpha\nu} + x_\nu\delta_{\alpha\mu}) + 4e^{\Omega}[\Omega'' + (\Omega')^2]x_\alpha x_\mu x_\nu, \tag{B.15}$$

which is symmetric under permutation of its indices. Then, we may use the simplified formula for the $(4, 0)$ Riemann tensor obtained in Appendix A, which yields

$$R_{\mu\nu\alpha\beta} = -\frac{3e^{\Omega}(\Omega')^2}{1 + 2\|\mathbf{x}\|^2\Omega'}\left[\|\mathbf{x}\|^2\delta_{\mu\alpha}\delta_{\nu\beta} + \left(1 + 2\|\mathbf{x}\|^2\frac{\Omega''}{\Omega'}\right)(x_\nu x_\beta\delta_{\mu\alpha} + x_\mu x_\alpha\delta_{\nu\beta}) - (\alpha \leftrightarrow \beta)\right] \tag{B.16}$$

after a straightforward computation, where we have noted that

$$g^{\rho\lambda}x_\rho = e^{-\Omega}\frac{1}{1+2\|\mathbf{x}\|^2\Omega'}x_\lambda \tag{B.17}$$

and

$$g^{\rho\lambda}x_\rho x_\lambda = e^{-\Omega}\frac{\|\mathbf{x}\|^2}{1+2\|\mathbf{x}\|^2\Omega'} \tag{B.18}$$

We can then compute the Ricci scalar

$$R = g^{\mu\alpha}g^{\nu\beta}R_{\mu\nu\alpha\beta} \tag{B.19}$$

$$= -\frac{3e^{\Omega}(\Omega')^2}{1+2\|\mathbf{x}\|^2\Omega'}\left[\|\mathbf{x}\|^2(g^{\alpha\alpha}g^{\beta\beta} - g^{\alpha\beta}g^{\beta\alpha})\right.$$

$$\left. + 2\left(1+2\|\mathbf{x}\|^2\frac{\Omega''}{\Omega'}\right)(g^{\alpha\alpha}g^{\nu\beta}x_\nu x_\beta - g^{\mu\alpha}g^{\mu\beta}x_\alpha x_\beta)\right] \tag{B.20}$$

which, as

$$g^{\alpha\alpha}g^{\beta\beta} - g^{\alpha\beta}g^{\beta\alpha} = e^{-2\Omega}\left(d - 2\frac{2\|\mathbf{x}\|^2\Omega'}{1+2\|\mathbf{x}\|^2\Omega'}\right)(d-1) \tag{B.21}$$

and

$$g^{\alpha\alpha}g^{\nu\beta}x_\nu x_\beta - g^{\mu\alpha}g^{\mu\beta}x_\alpha x_\beta = e^{-2\Omega}\frac{\|\mathbf{x}\|^2}{1+2\|\mathbf{x}\|^2\Omega'}(d-1) \tag{B.22}$$

yields

$$R = -\frac{3(d-1)e^{-\Omega}(\Omega')^2\|\mathbf{x}\|^2}{(1+2\|\mathbf{x}\|^2\Omega')^2}\left[d+2+2\|\mathbf{x}\|^2\left((d-2)\Omega'+2\frac{\Omega''}{\Omega'}\right)\right]. \tag{B.23}$$

The relation between Gaussian norms of derivatives of the activation function and input discriminability indicated by this result is consistent with previous studies of how the activation function affect the expressivity of infinite two-layer networks (Poole et al., 2016; Daniely et al., 2016; Zavatone-Veth and Pehlevan, 2022, 2021).

## B.2. Examples

As an analytically-tractable example, we consider the error function $\phi(x) = \mathrm{erf}(x/\sqrt{2})$. For such networks, the NNGP kernel is

$$k(\mathbf{x},\mathbf{y}) = \frac{2}{\pi}\arcsin\frac{\sigma^2\mathbf{x}\cdot\mathbf{y}+\zeta^2}{\sqrt{(1+\sigma^2\|\mathbf{x}\|^2+\zeta^2)(1+\sigma^2\|\mathbf{y}\|^2+\zeta^2)}}, \tag{B.24}$$

which is easy to prove using the integral representation of the error function (Saad and Solla, 1995). In this case, we have the simple result $\phi'(x) = \sqrt{2/\pi}\exp(-x^2/2)$, hence we can easily compute

$$\mathbb{E}[\phi'(z)^2] = \frac{2}{\pi\sqrt{1+2(\sigma^2\|\mathbf{x}\|^2+\zeta^2)}}. \tag{B.25}$$

This yields

$$\Omega(\|\mathbf{x}\|^2) = -\frac{1}{2}\log[1 + 2(\sigma^2\|\mathbf{x}\|^2 + \zeta^2)] + \log\frac{2\sigma^2}{\pi} \tag{B.26}$$

hence we easily obtain the volume element

$$\sqrt{\det g} = \left(\frac{2\sigma^2}{\pi}\right)^{d/2} \frac{\sqrt{2\zeta^2 + 1}}{[1 + 2(\sigma^2\|\mathbf{x}\|^2 + \zeta^2)]^{(d-2)/4}} \tag{B.27}$$

and the Ricci scalar

$$R = -\frac{3\pi(d-1)(d+2)\sigma^2\|\mathbf{x}\|^2}{2(2\zeta^2 + 1)\sqrt{1 + 2(\sigma^2\|\mathbf{x}\|^2 + \zeta^2)}}. \tag{B.28}$$

In this case, it is easy to see that $R$ is negative for all $d > 1$ and that it is a monotonically decreasing function of $\|\mathbf{x}\|$, hence curvature becomes increasingly negative with increasing radius.

Another illustrative example is the monomial $\phi(x) = x^q/\sqrt{(2q-1)!!}$ for integer $q \geq 1$, normalized such that

$$k(\mathbf{x}, \mathbf{x}) = \mathbb{E}[\phi(z)^2] = (\sigma^2\|\mathbf{x}\|^2 + \zeta^2)^q. \tag{B.29}$$

We remark that the resulting NNGP kernel will not in general simply be a polynomial kernel $(\mathbf{x} \cdot \mathbf{y})^q$, as it will include terms that depend on $(\mathbf{x} \cdot \mathbf{y})^{q-2}$, $(\mathbf{x} \cdot \mathbf{y})^{q-4}$, et cetera. An explicit formula for the NNGP kernel for two distinct inputs could in principle be obtained using the Mehler expansion of the bivariate Gaussian density (Daniely et al., 2016; Zavatone-Veth and Pehlevan, 2021), but we will not do so here. For these activation functions, we have

$$\mathbb{E}[\phi'(z)^2] = \frac{q^2}{2q-1}(\sigma^2\|\mathbf{x}\|^2 + \zeta^2)^{q-1}, \tag{B.30}$$

yielding the volume element

$$\sqrt{\det g} = \sqrt{1 + 2(q-1)\frac{\sigma^2\|\mathbf{x}\|^2}{\sigma^2\|\mathbf{x}\|^2 + \zeta^2}}\left(\frac{q^2\sigma^2(\sigma^2\|\mathbf{x}\|^2 + \zeta^2)^{q-1}}{2q-1}\right)^{d/2} \tag{B.31}$$

and the Ricci scalar

$$R = -\frac{3(d-1)(q-1)^2(2q-1)\sigma^2\|\mathbf{x}\|^2[(d+2)\zeta^2 + (d-2)(2q-1)\sigma^2\|\mathbf{x}\|^2]}{q^2(\sigma^2\|\mathbf{x}\|^2 + \zeta^2)^q[(2q-1)\sigma^2\|\mathbf{x}\|^2 + \zeta^2]^2}. \tag{B.32}$$

If $\zeta = 0$, this simplifies substantially to

$$R\bigg|_{\zeta=0} = -\frac{3(d-1)(d-2)(q-1)^2}{q^2(\sigma^2\|\mathbf{x}\|^2)^q}. \tag{B.33}$$

In this case, $R < 0$ for all $d > 2$, but, unlike for the error function, $|R|$ is monotonically decreasing with $\|\mathbf{x}\|$.

## Appendix C. Numerical methods and supplemental figures

As an especially simple toy problem, we begin by training neural networks to perform a standard XOR classification task. Single-hidden-layer fully-connected networks with either sigmoid or Softplus nonlinearities are initialized with widths $[2, 2, 1]$ and trained on a dataset consisting of the four points

$$\{(-1, -1), (-1, 1), (1, -1), (1, 1)\} \tag{C.1}$$

with respective labels $\{0, 1, 1, 0\}$. Networks are trained via stochastic gradient descent (learning rate 0.02, momentum 0.9, and weight decay $10^{-4}$) with mean-squared error loss for 2000 epochs.

For a slightly more complex toy problem, we train neural networks to classify points according to a sinusoidal decision boundary. Two-hidden-layer fully-connected networks are initialized with widths [2,8,8,2] and trained on a dataset consisting of 400 points $(x_1, x_2) \in [-1, 1] \times [-1, 1]$ with labels

$$y(x) = \begin{cases} 1 & x_2 > \frac{3}{5} \sin(7x_1 - 1) \\ 0 & x_2 < \frac{3}{5} \sin(7x_1 - 1) \end{cases} \tag{C.2}$$

Networks are trained via stochastic gradient descent (learning rate 0.05, momentum 0.9, and zero weight decay) with cross-entropy loss for 150,000 epochs.

In both cases, we calculate the volume element and Ricci scalar induced by the network at 1,600 points evenly spaced on a grid in $[-1.5, 1.5] \times [-1.5, 1.5]$ throughout training (the magnitudes of these two quantities at each of the 1,600 points are plotted as heat maps in Figures 1 and 2). The metric we consider is the one induced by the map from input space to the first hidden layer of the network (in the case of XOR, the single hidden layer). We compute this metric and the resulting curvature quantities using the equations listed in the main text, (2) and (4). All of the required derivatives with respect to input components are computed with automatic differentiation in PyTorch (Paszke et al., 2019).

Finally, we compute the metric induced on input space by networks trained to classify MNIST digits. Fully-connected networks with a single hidden layer of 30 nodes are trained on the Scikit-learn $8 \times 8$ pixel handwritten digit image dataset (Pedregosa et al., 2011). Batches of twelve images and their labels (numbers $0 - 9$) are fed to the network for 30 epochs; the networks are trained via the Adam optimizer (learning rate 0.005, weight decay $10^{-1}$) with negative log-likelihood loss. The metric induced by the trained network at a series of input images is then computed with autograd as described above. The images we consider are either drawn from the dataset, or are images $\mathbf{y}_i$ interpolated between two dataset images $\mathbf{x}_1$ and $\mathbf{x}_2$ as follows:

$$\mathbf{y}_j = \mathbf{x}_1 + \frac{t}{64}(\mathbf{x}_2 - \mathbf{x}_1) \tag{C.3}$$

for $t \in [0, 64)$. Eigenvalues of the metric matrix $g_{\mu\nu}$ tend to become small as training progresses, and so, due to the high dimensionality of the input space, the metric $\sqrt{\det g_{\mu\nu}}$

becomes minuscule and difficult to compute within machine precision. Therefore, instead of $\sqrt{\det g_{\mu\nu}}$, we compute an effective volume element from $g_{\mu\nu}$: the square-root of the product of the largest six eigenvalues of the metric matrix. We find that this effective volume element consistently grows (relatively) large at input images near the decision boundary, as shown in Figure 3.

