# OpenReview forum: "Training shapes the curvature of shallow neural network representations"
_NeurIPS.cc/2022/Workshop/NeurReps — NeurReps 2022 Poster_

### Official Review · Reviewer_Conx · 2022-10-12
**Concise well written paper on the behavior of curvature when training neural networks**

**Confidence:** 3
**Soundness:** 3
**Presentation:** 3
**Contribution:** 3
**Overall Rating:** 7

**Summary:**

This papers studies Riemannian properties (curvature and volume element) of the metric induced on the manifold defined as the image of a subset of the input space by the neural network. The results from the literature that ground this study are recalled, and some of the computations are detailed in appendix. These concern first finitely wide neural networks. In this case symmetries of the metric allow explicit computations of the curvature tensor and Ricci scalar. Then, infinitely wide networks are considered, and again the Ricci scalar can be computed thanks to spherical symmetry. Finally, numerical results from toy datasets are given and show that the volume element grows near boundary decisions while curvature decreases.

**Questions:**

The authors consider smooth activation functions. What could be done for RELU functions? As the behavior of neural networks is similar with both smooth and non smooth activation functions, what tools could we use to analyse jointly both cases?
The title and abstract are not clear: the verb "to shape" is hard to understand in this context if the reader is more familiar with shape analysis. Maybe "to model" would be more understandable? I only understood the meaning of the title after reading the end of the introduction.
As the color map changes, it is hard to see that the volume element actually increases on Figure 1, maybe keeping the color map fixed for the three images could make it more obvious?

**Limitations:**

The experiments are performed on rather small learning tasks.
Moreover, the analysis and computations of curvature are restricted to shallow networks, whereas the real challenge is to understand the benefits of depth, as this is the key to the revolution in machine learning.

**Recommended Decision:**

3: Accept

**Relevance:**

3: Solid fit

**Strengths And Weaknesses:**

The paper is well written and gives a concise but clear presentation of the necessary material as well as the contributions. The results seem well proven by the numerical experiments but maybe more interpretation/discussion could give a better grasp of the significance of these results.


**Submission Track:**

Extended Abstract (4 Page)

---

> ### Author Response · Authors · 2022-11-01
> **Response to Reviewer Conx**
>
> We thank the referee for their careful assessment of our abstract. In regards to their questions, within a single activation cell of a ReLU network, the feature map is linear, and therefore yields a flat metric. However, the non-differentiability of ReLU at the origin means that the signal manifold will not be globally smooth, and therefore not immediately amenable to study with the same tools used here. We defer further study of ReLU networks to future work. We thank the referee for their comment regarding the usage of "to shape" in the title, but we believe that its meaning is sufficiently clear from the context of the abstract. Finally, we note that studying deeper networks is more technically challenging, and is likewise the object of ongoing work.

---

### Official Review · Reviewer_UMVK · 2022-10-14
**Excellent paper with significant impact to understand the theory of neural networks using geometry.**

**Confidence:** 4
**Soundness:** 4
**Presentation:** 4
**Contribution:** 4
**Overall Rating:** 8

**Summary:**

The paper studies the geometry of infinite-width shallow neural networks, in particular how classification tasks impacts the curvature and the volume element of the Riemannian metric pulled-back from the feature space to the input space. With some assumptions, closed-form expressions are obtained for the metric, the Ricci scalar and the volume elements. The theoretical results are confirmed empirically.

**Questions:**

(1) The feature map is divided by the number of weights in the hidden layer: is it really correct in theory? In general, the feature map are only defined as $\phi(x) = w^{\top} x + b$, I understand that dividing by $n$ would ultimately help proving the central limit theorem. Yet, would you reach the same conclusion without dividing the metric by $n$? If not, can you justify the division?

(2) The number of features is defined to be always greater than the number of inputs: what could potentially happen in cases where $n \geq d$?

(3) Will the authors consider extending their work to non-shallow neural networks? Would the results likely be influenced by increasing the number of layers in a neural network?

**Limitations:**

No limitations to note

**Recommended Decision:**

3: Accept

**Relevance:**

4: Highly relevant

**Strengths And Weaknesses:**

The paper uses Riemannian geometry to better understand the training of shallow neural networks. It is both novel and highly significant for geometers and machine learners. It is is written clearly with well designed experiments. Only minors details could be improved such as justifying the definition of the metric, or explaining the soundness of some hypothesis.

**Submission Track:**

Extended Abstract (4 Page)

---

> ### Author Response · Authors · 2022-11-01
> **Response to Reviewer UMVK**
>
> We are gratified by the reviewer's favorable assessment of our abstract. Below, we provide a brief reply to their questions.
>
> 1. Our choice of scaling in the feature map follows what is known as Neural Tangent Kernel parameterization (c.f. Jacot et al., NeurIPS 2018 or Yang & Hu, ICML 2021), which ensures that the infinite-width limit of gradient descent training is well-defined. At the modest widths we tested in our preliminary experiments, qualitatively similar results hold with standard parameterization.
>
> 2. If the number of features is less than the number of inputs, then the feature space representation is generally no longer an embedding of the input manifold, and information is lost. We have added a comment regarding this point, which may be interesting to study in future work.
>
> 3. Extending these ideas to non-shallow networks is the object of ongoing work.

---

### Official Review · Reviewer_jb1g · 2022-10-15
**A valuable contribution toward understanding the Riemmanian geometry induced by neural networks.**

**Confidence:** 4
**Soundness:** 4
**Presentation:** 3
**Contribution:** 4
**Overall Rating:** 7

**Summary:**

There is growing interest in a detailed theoretical understanding of how the Riemannian geometry induced by neural networks affects learning. It is intuitive that local amplification of areas (as represented by the volume element) and a local flattening of the space (quantified by the Ricci curvature scalar) in the vicinity of a decision boundary would improve discrimination tasks. This work aims to characterize how these geometric quantities of the representational manifold change over training.

**Questions:**

Suggestions: There are a few minor typos throughout the text, and the figures need to be improved for readability. Figure 3 in particular is quite messy, and should be reworked.

**Limitations:**

As previously mentioned, since the study claims to characterize "how" training shapes the Riemannian curvature, there should be some mention of the convergence toward the final state. Importantly, there should also be an attempt to estimate the statistical error.

**Recommended Decision:**

3: Accept

**Relevance:**

4: Highly relevant

**Strengths And Weaknesses:**

This extended abstract makes a valuable contribution to the topic. Comprehensively, I consider this a strong paper that is well presented and makes a persuasive argument for expanding upon this work in the future. The appendix provides a thorough analytic derivation of the relevant geometric quantities. One contribution of value is the generalization of the spherical symmetry of the Ricci scalar for infinite-width shallow networks to arbitrary (smooth) activation functions. This paper makes a strong argument that the training of neural networks for simple tasks indeed locally expands areas and reduces curvature near decision boundaries. One weakness is there is no mention of temporal dynamics of the Riemannian geometry throughout learning, or a quantitative characterization of the convergence toward the final final state.

**Submission Track:**

Extended Abstract (4 Page)

---

> ### Author Response · Authors · 2022-11-01
> **Response to reviewer jb1g**
>
> We thank the referee for their kind comments and favorable assessment of our work. Their suggestions regarding the readability of the figures and the need to quantify convergence are well-taken, and will be taken into account as we prepare a longer-form version of this work.

---

### Decision · Program_Chairs · 2022-10-21

Accept (Poster)